

# On the relationship between tumour growth rate and survival in non-small cell lung cancer

Hitesh B. Mistry

Division of Pharmacy, University of Manchester, Manchester, United Kingdom

## ABSTRACT

A recurrent question within oncology drug development is predicting phase III outcome for a new treatment using early clinical data. One approach to tackle this problem has been to derive metrics from mathematical models that describe tumour size dynamics termed re-growth rate and time to tumour re-growth. They have shown to be strong predictors of overall survival in numerous studies but there is debate about how these metrics are derived and if they are more predictive than empirical end-points. This work explores the issues raised in using model-derived metric as predictors for survival analyses. Re-growth rate and time to tumour re-growth were calculated for three large clinical studies by forward and reverse alignment. The latter involves re-aligning patients to their time of progression. Hence, it accounts for the time taken to estimate re-growth rate and time to tumour re-growth but also assesses if these predictors correlate to survival from the time of progression. I found that neither re-growth rate nor time to tumour re-growth correlated to survival using reverse alignment. This suggests that the dynamics of tumours up until disease progression has no relationship to survival post progression. For prediction of a phase III trial I found the metrics performed no better than empirical end-points. These results highlight that care must be taken when relating dynamics of tumour imaging to survival and that bench-marking new approaches to existing ones is essential.

## INTRODUCTION

A key question being posed in early oncology drug development at the end of phase II (and increasingly at the end of phase I expansion studies) is: given past imaging information, what is the overall survival (OS) outcome in a phase III study likely to be? This question has been primarily addressed by analysing efficacy data using RECIST criteria (Response Evaluation Criteria In Solid Tumours) (*Therasse et al., 2000*; *Eisenhauer et al., 2009*). RECIST v1.0 and v1.1 involves the evaluation of a drug's efficacy against both target and non-target lesions. The drug's effect on target lesions are reported via the Sum of Longest Diameters (SLD) marker which is a continuous variable whereas drug effect on non-target lesions is reported via qualitative descriptions: increase, decrease or no-change. Both are recorded over time as is the occurrence of a new lesion. The variables discussed above can also be combined to produce one variable, RECIST response classification which consists

Corresponding author
Hitesh B. Mistry,
hitesh.mistry@manchester.ac.uk

of four categories: Complete Response (CR), Partial Response (PR), Stable Disease (SD) or Progressive Disease (PD). CR represents complete disappearance of both target and non-target lesions. PR and SD are simply categorised versions of percentage change in SLD. PD as well as being a categorised version of percentage change in SLD also considers increase in non-target lesion size and occurrence of a new lesion. An important continuous metric of interest is progression free survival (PFS) time: the time from treatment initiation to PD or death. This time-point is of importance for two reasons; firstly it marks the end of imaging data collection and secondly, it also signals the end of the current treatment and the beginning of the next. Therefore, the imaging data collected can only provide one with information on the effect of the first treatment given during the trial. Without knowing what effect the treatment post progression has on the disease clearly makes survival prediction challenging. However, if one assumes the subsequent treatments have minimal effect on disease, compared to initial treatment, then survival prediction may be possible. The debate then moves onto how best to analyse the imaging data collected up until disease progression.

The option of using continuous measures involving changes in SLD over the categorisation scheme has been a long-standing debate when analysing early patient response to treatment (*An et al., 2011*; *An et al., 2015*; *Toffart et al., 2014*). These works have shown that there was no benefit in terms of survival concordance probability when using continuous changes in SLD compared to the current RECIST response classification. More elaborated approaches have therefore explored the use of mathematical models to fit the time-series of SLD (*Stein et al., 2008*; *Stein et al., 2009*; *Stein et al., 2012*; *Wang et al., 2009*; *Blagoev et al., 2013*; *Claret et al., 2013*; *Bruno, Mercier & Claret, 2014*; *Han et al., 2016*). Figure 1A shows the metrics that have been derived from these models of SLD time-series. These are the decay rate (DR), the re-growth rate (GR), the model estimated percentage change at a certain time-point and the time to tumour re-growth (TTG). In particular, TTG and GR have been claimed to be significant predictors of survival (*Stein et al., 2008*; *Stein et al., 2009*; *Stein et al., 2012*; *Blagoev et al., 2013*; *Claret et al., 2013*; *Han et al., 2016*) when calculated using an exponential decay/growth model. However, concerns about how these two metrics are estimated have been raised. The first relates to drop-outs not being accounted for through a joint longitudinal and survival model (*Ibrahim, Chu & Chen, 2010*; *Mansmann & Laubender, 2013*). The second concerns the time-dependent bias of TTG and GR, that is the time taken to estimate these quantities was never accounted for in those previous studies (*Van Walraven et al., 2004*; *Mistry, 2016*).

Contradictory results have been reported regarding the predictive power of these metrics. In one hand, *Sharma et al. (2015)* used TTG within a resampling of a single phase III study to assess phase II trial design and endpoints. They showed that TTG was equal if not superior to PFS in making the correct decision with regards to moving from phase II to III given an 18 week landmark time-point. On the other hand, *Kaiser (2013)* used a larger study and found that PFS was a superior endpoint compared to GR in five of his six trials. There are two possible reasons why the results from Sharma et al. and Kaiser differ: (i) the difference in PFS between treatments, in the studies used by Kaiser, were greater than

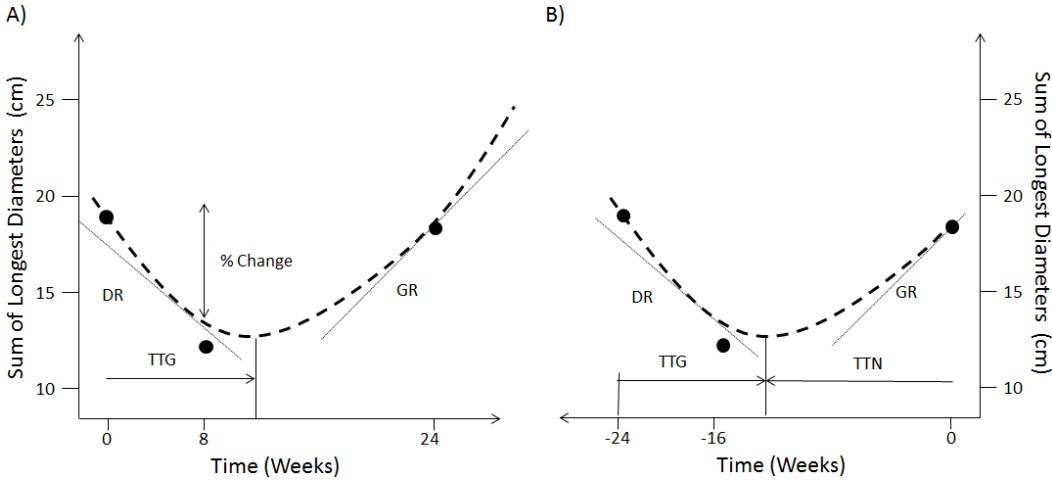

**Figure 1  Forward and reverse alignment of SLD time-series.** (A) Different model derived metrics, % change at week 8, TTG, DR and GR are obtained by fitting a model (dashed line) to actual observations (dots) in forward time. (B) Same model derived metrics and observations in reverse time as well as time to tumour nadir (TTN).

that in the study used by Sharma et al. and/or (ii) the difference could be attributed to the choice of model based metric used: Sharma et al. used TTG whereas Kaiser chose GR.

It is important to note that within the context of this study, risk can be allocated within two different categories, individual and group risk. An individual risk metric is assessed through survival concordance probabilities (*Gönen & Heller, 2005*; *An et al., 2011*; *An et al., 2015*), whereas a group risk metric is assessed on whether the correct decision was made to advance a compound from phase II to phase III. In this context, individual risk estimates the patient's survival prognosis based on the patient's historical imaging time series data. Therefore, for individual risk, the time taken to acquire sufficient time-series to estimate GR and TTG must be accounted for otherwise one assumes one can see into the future. This was not done in the approaches taken in *Stein et al. (2008)*, *Stein et al. (2009)*, *Stein et al. (2012)*, *Blagoev et al. (2013)*, *Claret et al. (2013)* and *Han et al. (2016)* and so a question remains as to whether GR and TTG can be used to predict individual risk.

For group risk the question of interest is different: given historical data on a standard of care treatment in a group of patients, is a new treatment (which has mature PFS data in a group of patients) likely to produce survival improvement? This would be assessed for instance through a future prospective head-to-head trial. In this different context, full time courses can indeed be considered for metrics estimation as was the case in the approaches taken by *Stein et al. (2008)*, *Stein et al. (2009)*, *Stein et al. (2012)*, *Blagoev et al. (2013)*, *Claret et al. (2013)* and *Han et al. (2016)*. It could be argued that individual risk is mainly a concern to the practicing medical community whereas group risk is a concern for the pharmaceutical industry and regulatory agencies. However, given the growth of interest in personalised (precision) medicine, individual risk is also likely to become a critical factor of future drug development.

The objective of this study was to analyse both individual and group risk using three comparator arms of phase III NSCLC (Non-Small Cell Lung Cancer) studies in the following way. For individual risk the model derived metrics, GR and TTG, calculated using the exponential decay/growth model used in previous studies (*Stein et al., 2008*; *Stein et al., 2009*; *Stein et al., 2012*; *Kaiser, 2013*; *Blagoev et al., 2013*; *Claret et al., 2013*; *Sharma et al., 2015*), were assessed via calculation of the survival concordance probability. Concordance probability was chosen for two reasons: (1) it is a measure that is routinely used when reporting the predictive capability of a prognostic model developed for use in the clinic (*Gönen & Heller, 2005*; *An et al., 2011*; *An et al., 2015*); (2) it is a non-parametric approach which I preferred over a regression model to avoid the use of a specific functional form when measuring evidence of relationship between variables. I examined the correlation to overall survival times for these quantities using two different data alignment approaches, forward and reverse. Forward alignment refers to correlating TTG and GR to overall survival without accounting for the time taken to estimate them. Reverse alignment involves re-aligning patients time-series to their time of progression and correlating TTG and GR to overall survival time minus the progression time. This approach not only accounts for the time taken to estimate GR and TTG but also assesses if those metrics are significant correlates to survival post progression. For group risk, I tested the ability of the metrics, PFS, TTG and GR, derived from the control arms of two of the phase III studies, to predict the outcome of a phase III study, where the treatments in the control arms were placed in a head-to-head trial, was tested.

## METHODS

### Data

Data from the control arm of three randomised phase III studies in NSCLC were collected: Erlotinib (*Scagliotti et al., 2012*), Docetaxel (*Ramlau et al., 2012*) and Paclitaxel/Carboplatin (*Socinski et al., 2012*). (Note, these studies were chosen as they were the only ones available at the time within ProjectDataSphere.) Both the Erlotinib and Docetaxel studies were conducted within a patient population who were pre-treated with doublet chemotherapy (majority had Paclitaxel/Carboplatin) whereas the Paclitaxel/Carboplatin study was conducted in therapy naïve patients. All three comparator arms were from studies designed to assess the overall survival of a new investigational treatment. Tumour assessments were conducted every eight weeks within the Docetaxel and Paclitaxel/Carboplatin studies and every six weeks in the Erlotinib study using RECIST 1.0. In order to calculate model derived metrics only patients that had a pre-treatment and at least one on-treatment tumour assessment were considered.

### Time-series drop-out mechanism

The protocols of the studies under consideration here stated that imaging data would cease to be collected once a patient's disease had progressed. Since death is considered as a progression event I assessed what proportion of progression events were due to death. The result of this analysis determined whether or not a joint longitudinal survival analysis was performed. If the predominant drop-out mechanism was due to reasons other than death

then a joint longitudinal survival model was not considered as the drop-out mechanism is then not informative of survival.

## Forward v reverse alignment

In reverse alignment the SLD measurement at the time of progression becomes time 0 and the first measurement becomes -$t$ days, see Fig. 1B. Furthermore, in forwards alignment I assessed if TTG and GR correlated to overall survival. But in reverse alignment I assessed if TTG and GR correlated to overall survival minus progression free survival. Therefore in forwards alignment I shall consider TTG and GR to be biased (time-dependent bias) and in reverse alignment un-biased.

## Model

In order to make the results comparable to other publications within the field (*Stein et al., 2008*; *Stein et al., 2009*; *Stein et al., 2012*; *Kaiser, 2013*; *Blagoev et al., 2013*; *Claret et al., 2013*; *Sharma et al., 2015*) I used the same decay and re-growth model as those studies to analyse the SLD time-series. The SLD is modelled as:

$$SLD(time) = A * (\exp(-B * time) + \exp(C * time) - 1).$$

The model was placed within a mixed model framework with an additive residual error model and parameters A, B and C considered to be log-normally distributed, as was done previously (*Kaiser, 2013*; *Claret et al., 2013*; *Sharma et al., 2015*). The turning point time (tp) was calculated by taking the derivative of SLD (time) with respect to time and setting it equal to 0, giving:

$$tp = (\log(B) - \log(C))/(B + C).$$

In forward alignment this is referred to as time to tumour re-growth (TTG; Fig. 1A); in reverse alignment it is the time to nadir (TTN) from the last observation (Fig. 1B). To convert TTN to TTG, TTN is subtracted from the total imaging observation time. In forwards time B and C will be DR and GR respectively, whereas it is the other way around in reverse time. The non-linear mixed model analysis was conducted using the *nlme* package in R v3.1.1 (*Pinheiro et al., 2017*). Details on how the model fits to the time-series data can be found in (Figs. S1–S6 contain diagnostic plots whereas Tables S1–S6 contain parameter values).

## Individual risk analysis

The model derived metrics GR and TTG across the three studies were assessed for their relationship to survival via an analysis of their concordance probability estimates. The concordance probability (CP) represents the probability that for any pair of patients, the patient with the better covariate value has the longer survival time. A covariate was called significant if its 95% bootstrapped (1,000 samples) concordance probability (CP) confidence intervals did not include 0.5 (*Sedgwick, 2013*). A value of 0.5 represents the case where there is not a consistent relationship. Concordance probability estimates were generated using the *CPE* package in R v3.1.1 (*Mo, Gonen & Heller, 2012*).

**Table 1  Key information on the three randomised phase III studies analysed here.**

|  | Erlotinib | Docetaxel | Paclitaxel/Carboplatin |
|---|---|---|---|
| Line of therapy | 2nd | 2nd | 1st |
| Total $N$ | 369 | 399 | 413 |
| No. death events | 61 | 282 | 289 |
| No. progression events | (301) | (353) | (328) |
| (Death events) | (27) | (20) | (6) |
| Median baseline SLD (cm) | 8 | 8.3 | 10.7 |
| 25th–75th Percentile | 5.2, 12.5 | 5.1, 12 | 6.9, 15.5 |
| Median% change (weeks 6–10) | 4.9 | 0 | −16.5 |
| 25th–75th percentile | −6.6, −27.3 | −10.5, −5.5 | −29.8, −5.9 |
| Median PFS (months) | 3.7 | 4.2 | 6.7 |
| (95% CI) | (2.6–3.8) | (4.1–4.7) | (6.0–7.1) |
| Median OS (months) | NA | 10.8 | 12.1 |
| (95% CI) |  | (9.7–12.2) | (11.2–13.2) |
| Median OS-PFS (months) | NA | 5.2 | 4.1 |
| (95% CI) |  | (4.3–5.8) | (3.7–5) |

Notes.
SLD, Sum of Longest Diameters; PFS, Progression Free Survival; CI, Confidence Interval.

## Group risk analysis

Tumour size metrics for Erlotinib and Docetaxel study arms were used to predict the outcome of a head-to-head phase III (test) trial (*Garassino et al., 2013*). I applied the following methodology, as used by *Sharma et al. (2015)* and *Kaiser (2013)*, to generate distributions of OS HR using tumour size metrics. A total of 1,000 test trial data-sets were generated by sampling with replacement the exact same number of patients used in each study arm of the test trial ($n = 110$ for Docetaxel and $n = 112$ for Erlotinib). In generating 1,000 test trial data-sets, distributions of the following outcome measures were generated. Evaluation of TTG and GR metrics was done by calculating the ratio of the mean values between the two arms of the test trial, subsequently a distribution of these ratios were generated and for PFS the distribution of PFS HR was generated. The distribution of the predicted OS HR from TTG, GR and PFS was visualised using histograms with the actual results of the study with 95 percent confidence intervals overlaid. In addition to the visual inspection I reported the median OS HR prediction with 95 percent prediction intervals for each of the metrics.

# RESULTS

## Patient data and drop-out mechanism

Details of the patients' imaging and survival characteristics can be seen in Table 1. The table highlights that the number of progression events due to death were very low across all studies. Thus, the predominant reason why imaging time-series ceases to be collected is due to events other than death. The drop-out mechanism is therefore not informative of survival. Instead drop-out is informative of when the patient discontinues treatment.

**Table 2   Summary of concordance probability estimates of GR (Growth Rate) and TTG (Time to Tumour Re-Growth) metrics: forward versus reverse alignments.** Concordance probabilities (CP) and 95% confidence intervals (CI) are shown for GR and TTG metrics when using forwards vs. reverse alignment to analyse the time-series, for three randomised phase III studies.

|  | Erlotinib | | Docetaxel | | Docetaxel/carboplatin | |
|---|---|---|---|---|---|---|
|  | Forward CP (95% CI) | Reverse CP (95% CI) | Forward CP (95% CI) | Reverse CP (95% CI) | Forward CP (95% CI) | Reverse CP (95% CI) |
| GR | **0.63 (0.55–0.71)** | 0.49 (0.43–0.59) | **0.69 (0.65–0.72)** | 0.52 (0.49–0.56) | **0.71 (0.68–0.74)** | 0.53 (0.48–0.57) |
| TTG | **0.64 (0.57–0.71)** | 0.53 (0.47–0.61) | **0.70 (0.66–0.73)** | 0.54 (0.48–0.58) | **0.72 (0.68–0.75)** | 0.55 (0.49–0.59) |

## Individual risk

Table 2 shows that the alignment method affects CP values. I found that GR and TTG are strong covariates for overall survival across all treatments using forward alignment; where time taken to estimate TTG and GR was not accounted for. However, I found they did not correlate to overall survival minus progression free survival, reverse alignment. These results show that GR and TTG provide no information on the survival prognosis of patients post progression.

## Group risk

I then attempted to predict the outcome of the phase III test trial of Erlotinib versus Docetaxel, which showed an advantage of Docetaxel over Erlotinib, using GR, TTG or PFS. It can be seen in the histograms in Fig. 2A that the predicted distribution of OS HR using either GR (green) or PFS (grey) are close to the observed OS HR (solid vertical line) and sit well within the observed 95 percent confidence intervals (dashed vertical lines). This however is not the case for the OS HR predicted using TTG (pink) whose predicted distribution appears to sit outside the observed confidence intervals. The median and 95 percent prediction interval for the OS HR using each of the metrics, PFS, GR and TTG, were, 0.84 (0.62–1.13), 0.64 (0.53–0.77) and 0.42 (0.25–0.62) respectively. In comparing these to the actual OS HR observed, 0.73 (0.53–1), it is clear that the OS HR predicted using TTG is the furthest away. Both PFS and GR however are equally as close to the actual results with PFS under-estimating the effect and GR over-estimating the effect.

In order to investigate the difference between the predicted OS HR distributions of TTG and GR I assessed the correlation between the ratios of means between the two metrics, see Fig. 2B. The correlation plot shows that there is a relationship between the two variables ($r^2 = 0.36$) but that it is weak. This suggests that the information held within GR and TTG is moderately different and that the choice of metric could lead to different predictions. This may explain why TTG and GR did not give comparable predictions.

## DISCUSSION

In this work I examined the relationship between on-treatment changes in imaging and survival outcomes, which has been gaining favour recently (*Stein et al., 2008*; *Stein et al., 2009*; *Stein et al., 2012*; *Wang et al., 2009*; *Blagoev et al., 2013*; *Claret et al., 2013*; *Bruno, Mercier & Claret, 2014*; *Han et al., 2016*). Based on the cases I have analysed, I found that:

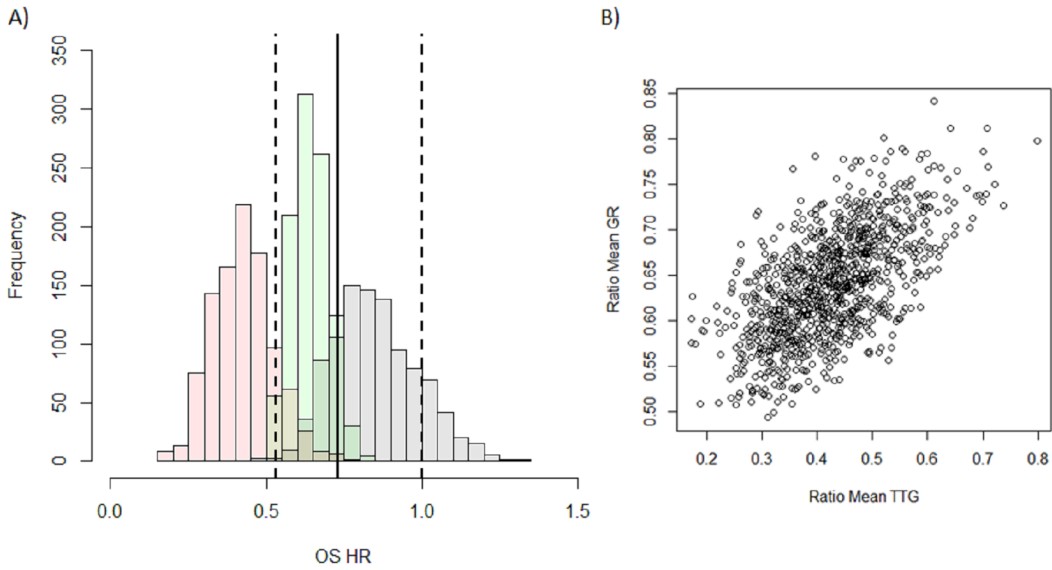

**Figure 2** **Distribution of model-based covariates.** (A) Histograms showing the predicted OS HR distribution for PFS (grey), GR (green) and TTG (pink) obtained from the 1,000 samples of the test trial using historical data. The OS HR of the actual results of the study are overlaid: solid vertical lines is the point estimate of the HR with dashed lines being the 95 percent confidence intervals. (B) Correlation between the ratio of mean TTG versus ratio of mean GR between each arm for each test trial.

(1) Model derived metrics TTG and GR were not significant covariates of survival post progression.

(2) The outcome of the phase III study using historical data was predicted just as well by conventional PFS HR as TTG and GR.

Regarding the first point, model derived metrics TTG and GR were found to be significant correlates for overall survival when the time taken to estimate them was not accounted for i.e., they were considered to be biased. However, by re-phrasing the question to, do TTG and GR correlate to survival post progression? I accounted for this bias, since imaging data ceases to be collected once a patients disease progresses. The result of re-phrasing the question highlighted that the dynamics of the disease leading up to progression have no bearing on the prognosis of a patient post progression. This result may seem surprising however it may not be once one considers what happens at the time of progression. The patient stops taking the current line of treatment and at some time-point post-progression may start a new treatment. Furthermore, this new treatment may be quite variable if there is no standard treatment approved. Therefore unless there is a correlation between the dynamics leading up to progression and the dynamics under the new unknown treatment one would expect the results found here to hold in most clinical trials.

Regarding the second finding, I assessed three metrics in their ability to correctly predict the outcome of a phase III trial, an assessment of group risk, which was a head-to-head test of two of the drugs used in this analysis, Erlotinib and Docetaxel. The three metrics assessed were PFS HR, TTG and GR. Both PFS and GR were as far away to the actual result as each other; with GR over-predicting and PFS under-predicting the advantage of Docetaxel over

Erlotinib. TTG, however, was furthest away from the actual result; it overestimated the advantage of Docetaxel over Erlotinib more so than GR. These results confirm previous findings where PFS was equal or superior to GR in making the correct decision in moving from phase II to phase III (*Kaiser, 2013*). This is in contrast with the study by *Sharma et al. (2015)* where PFS was found to be slightly less predictive than TTG. A reason for this could be that the study used here as the test case was powered to show a PFS difference which was not the case in the study done by Sharma et al., where they stopped the re-sampled study after 18 weeks and so was not powered to detect a PFS difference which is usually done for phase II studies. Sharma et al. proposed that the difference in magnitude of PFS between their study arms and that of Kaiser explained why the latter found PFS superior to GR. However, the PFS HR I investigated here (0.71) was similar to the one in the study by Sharma et al. (HR: 0.79). I suggest that the reason for the difference in results between Sharma et al. and Kaiser is related to the choice of model derived metric. Sharma et al. used TTG whereas Kaiser et al. used GR. The results here show that the ratio of means of those metrics between treatment arms do not correlate strongly. Therefore, the discrepancy could be due to choice of metric and not to differences in PFS. It is interesting to note that although TTG and GR were not strong covariates for survival post-progression, they performed well at predicting group risk. Thus, it is possible that a different approach should be used for modelling individual and group risk.

Although the analyses performed were for only one disease type, NSCLC, and were within a limited pharmacological space, they highlight the importance of accounting for the time it takes to estimate model derived metrics when assessing their prognostic value for individual risk. In addition to the approach described here regarding re-aligning a patients time-series to their time of progression an alternative approach would be to perform a landmark analysis in forward alignment at a specific time-point as was done by *Sharma et al. (2015)*. Both of these approaches account for the time taken to estimate TTG and GR. For group risk, although the results here show that PFS HR can predict OS HR, it has been shown that in general this may not be the case (*Blumenthal et al., 2015*). However, this does not suggest that PFS cannot be used for decision making as I found that for majority of patients radiological progression precedes death; it merely implies that it may not always predict quantitatively what the OS difference could be.

Overall, when comparing the results seen here with previous published studies assessing the decision making value of tumour size metrics, it suggests that PFS remains the best metric to use for decision making when considering group risk. For individual risk, model derived metrics derived from SLD time-series might be inappropriate. This is in contrast to prognostic models developed using standard pre-treatment clinical variables (e.g., lactate dehydrogenase, alkaline phosphatase etc.) which have been shown to be predictive for a specific drug within a specific disease setting (*Halabi et al., 2014*; *Wendling et al., 2016*). It would be interesting to further investigate how prognostic models developed using routine clinical variables could be adapted to be used for predicting group risk and how well these perform compared to PFS for decision making and quantitatively predicting treatment differences in OS.

## ACKNOWLEDGEMENTS

The author wishes to thank Professor Leon Aarons, Dr. Fernando Ortega, Dr. Giovanni Di Veroli and Dr James Yates for helpful discussions. This publication is based on research using information obtained from http://www.projectdatasphere.org, which is maintained by Project Data Sphere, LLC. Neither Project Data Sphere, LLC nor the owner(s) of any information from the web site have contributed to, approved or are in any way responsible for the contents of this publication.

### Funding

Hitesh B. Mistry was funded by Manchester School of Pharmacy through an AstraZeneca grant. The funders had no role in study design, data collection and analysis, decision to publish, or preparation of the manuscript.

### Grant Disclosures

The following grant information was disclosed by the author:
Manchester School of Pharmacy.

### Competing Interests

The authors declare there are no competing interests.

### Author Contributions

- Hitesh B. Mistry conceived and designed the experiments, performed the experiments, analyzed the data, contributed reagents/materials/analysis tools, wrote the paper, prepared figures and/or tables, reviewed drafts of the paper.

### Data Availability

The data is available through the ProjectDataSphere platform, which doesn't allow re-publication of their data.

https://www.projectdatasphere.org/projectdatasphere/html/home.

All users have access to the data after registration: The 3 data-sets of interest used were:

https://www.projectdatasphere.org/projectdatasphere/html/content/133;

https://www.projectdatasphere.org/projectdatasphere/html/content/115;

https://www.projectdatasphere.org/projectdatasphere/html/content/108.

### Supplemental Information

Supplemental information for this article can be found online at http://dx.doi.org/10.7717/peerj.4111#supplemental-information.

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
