# Peer review of "On the relationship between tumour growth rate and survival in non-small cell lung cancer"

_PeerJ, doi:10.7717/peerj.4111_

## Round 0.1 · original submission · Minor Revisions

This is an interesting manuscript and clearly explains some important methodological problems which are often ignored in analyses of this data. The layout and detail are appropriate and there are only minor issues raised by the peer reviewers.

The landmark approach is briefly mentioned with regards to the analysis of Sharma. It is interesting that this unbiased approach hasn’t been used here for the forward alignment analyses (to solve time-dependent bias line 156). Although it has been clearly stated that there are biases with the current approach you could add an explicit point regarding the landmark approach in the discussion.

It is commendable that a pre-print of this manuscript has been published (https://www.biorxiv.org/content/early/2017/05/17/109934). However it is unclear, exactly what role Dr Ortega played in this research and manuscript as he is listed as a co-author in that version but only acknowledged here? Please could you clarify this in your response? Depending on his input he should either be included as a co-author here or a note added to the acknowledgements to clarify the role he played (to explain why he is a co-author on the pre-print).

The manuscript also requires another edit. e.g. The reference is missing on line 164. The word ‘how’ is missing at the end of 173. Lines 150-151 are confusing – it may be clearer to drop them and start with “In reverse alignment…”

Reviewer 1 ·

Basic reporting

The text is well written in clear, professional English to a good standard.

Experimental design

Lines 133 - 140. This section would be improved if something could be said about these three trials. For example, were these trials in a primary or metastatic setting? Were patients previously treated? While there are references to the trials, it would aid the reader to have at least some basic background without having to look up three different references.

The trial referred to as Paclitaxel/Carboplatin (ref 23) does not appear to have used Paclitaxel - the trial was actually comparing pemetrexed-carboplatin with etoposide-carboplatin. The reference to Paclitaxel/Carboplatin is made in several places in the text and should be corrected throughout.

Additionally, there is no discussion of how these trials were selected. Were they the only three trials with data availability for the model, or were these three selected from a pool of candidate studies? If selected from a pool what were the selection criteria?

Validity of the findings

No comment.

Additional comments

An interesting question for the discussion is whether disease setting has an impact on these analyses? For example the pemetrexed/carboplatin trial was in chemo-naive patients, whereas the other two trials were in previously treated patients. Does it make sense to imagine that the disease trajectory would be the same in these two very different populations? Is it correct to assume that the same model would apply? Similarly, the point is made in the discussion that these results pertain to NSCLC - some discussion on whether these results generalise to other solid tumours would be helpful.

Reviewer 2 ·

Basic reporting

I found this to be a very interesting and well-written paper with appropriate references to the relevant literature. The paper was well-structured and cogently argued.

Experimental design

The paper addresses a very important question - how to predict phase III outcome for a
new anti-cancer treatment using early clinical data.

The statistical analyses carried out on the data (tumour re-growth rate, time to tumour re-growth and survival) were entirely appropriate and well-presented.

Validity of the findings

I found the findings compelling and therefore the conclusions of the statistical analysis are very important and have important implications for future design of phase III clinical trials for anti-cancer drugs.

Additional comments

This is a very sound paper with interesting findings which deserves to be published.

Some minor typographical errors which should be corrected are:

lines 51, 52: "the evaluation of a drugs’ efficacy..." -> "the evaluation of a drug's efficacy..."

line 52: "The drugs effect on target lesions are reported..." -> "The drug's effect on target lesions is reported..."

line 199: "Details of the patients imaging and survival characteristics..." -> "Details of the patients' imaging and survival characteristics..."

line 221: "it’s clear that..." -> "it is clear that..."

line 240, 241: "However, by re-phrasing the question to, do TTG and GR correlate to survival post progression, to..." -> "However, by re-phrasing the question to, do TTG and GR correlate to survival post progression?, to..."

---

## Round 0.2 · accepted · Accept

Thank you for making the required changes and for explaining the authorship discrepancy. We are pleased to accept the manuscript for publication.

Reviewer 1 ·

Basic reporting

Pass.

Experimental design

Pass.

Validity of the findings

Pass.

Additional comments

Thank you for addressing the various comments raised in the previous round. I am happy for this interesting paper to be published in its revised form.

Reviewer 2 ·

Basic reporting

All good.

Experimental design

All good.

Validity of the findings

All good.

Additional comments

All requested changes were made - the manuscript is ready for publication in my opinion.

Overall, a very timely, original and interesting piece of work.